# Genetic and pharmacologic inhibition of calcineurin reduces biofilm formation by the pathogenic fungus *Trichosporon asahii* in an *in vivo* silkworm infection model

Yasuhiko Matsumoto 📧*, Yuta Shimizu, Mei Nakayama, Mai Takizawa, Sanae Kurakado, Takashi Sugita

Department of Microbiology, Meiji Pharmaceutical University, 2-522-1, Noshio, Kiyose, Tokyo, Japan

* ymatsumoto@my-pharm.ac.jp

## Abstract

*Trichosporon asahii* is a dimorphic pathogenic fungus that causes catheter-related bloodstream infection in immunocompromised patients with neutropenia. Biofilm formation by *T. asahii* on the surfaces of medical devices such as catheters is influenced by various host environmental factors. Calcineurin, a protein phosphatase composed of the catalytic subunit Cna1 and the regulatory subunit Cnb1, regulates multiple stress responses and virulence of *T. asahii*. The role of calcineurin in biofilm formation under host-derived conditions, however, remains unclear. Here, we demonstrated that calcineurin is essential for biofilm formation *in vivo* by *T. asahii*. While the *cna1* gene- and the *cnb1* gene-deficient mutants formed biofilms comparable to those of the parent strain *in vitro*, it produced significantly less biofilm than the parent strain in the *in vivo* silkworm infection model. Similarly, tacrolimus, a calcineurin inhibitor, did not inhibit biofilm formation by *T. asahii in vitro* but markedly suppressed biofilm formation *in vivo*. Together, these findings suggest that calcineurin plays a crucial role in biofilm formation by *T. asahii* under host environmental conditions.

## Introduction

*Trichosporon asahii* is a pathogenic fungus with multiple morphologic forms, including yeast cells, hyphae, and arthroconidia [1]. In immunocompromised patients with neutropenia, catheter-related fungemia is a severe bloodstream infection caused by *T. asahii* biofilm formation on the catheter surface. [2,3]. The *T. asahii* biofilm comprises fungal cells embedded in an extracellular matrix composed of polysaccharides, proteins, nucleic acids, and lipids [4,5]. Biofilm-associated *T. asahii* exhibits resistance to ethanol and antifungal drugs, such as amphotericin B, caspofungin, and voriconazole [4,6]. Therefore, elucidating the molecular mechanisms that govern biofilm formation by *T. asahii* is essential for developing effective preventive strategies against catheter-related bloodstream infections.

of the Creative Commons Attribution License, which permits unrestricted use, distribution, and reproduction in any medium, provided the original author and source are credited.

**Data availability statement:** All relevant data are within the paper and its Supporting Information files.

**Funding:** This study was financially supported by the Japan Society for the Promotion of Science (JSPS) in the form of grants received by YM (JP23K06141) and SK (JP25K18621). This study was also financially supported in part by the Research Program on Emerging and Re-emerging Infectious Diseases of the Japan Agency for Medical Research and Development, AMED, in the form of a grant received by TS (JP24fk0108679h0402). The funders had no role in study design, data collection and analysis, decision to publish, or preparation of the manuscript. No additional external funding was received for this study.

**Competing interests:** The authors have declared that no competing interests exist.

Because biofilm formation by pathogenic fungi on catheter surfaces is influenced by host environmental factors, such as nutrients, host proteins, and interactions with host cells [7–9], evaluating these interactions *in vivo* is essential to elucidate the mechanisms of fungal biofilm formation under host-derived conditions [9]. The catheter-inserted silkworm infection model, in which a single polyurethane fiber–the same material used for medical catheters–is inserted into the silkworm hemolymph under the skin surface, has been used to investigate biofilm formation by pathogenic fungi *in vivo* [10–12]. For example, some studies using this model demonstrated that *Candida albicans* forms biofilms on the fiber surface in the hemolymph [10,13] and exhibits tolerance to antifungal agents, including amphotericin B, fluconazole, and voriconazole [10,11]. In another recent study, biofilm formation by *T. asahii* in this model was shown to be regulated by the mitogen-activated protein kinase Hog1 [12]. These studies demonstrate that the catheter-inserted silkworm infection model is a valuable system for evaluating *T. asahii* biofilm formation *in vivo*.

Calcineurin is a highly conserved serine/threonine-specific, $Ca^{2+}$/calmodulin-activated protein phosphatase in eukaryotes, including fungi [14,15]. Calcineurin forms a heterodimeric complex composed of the catalytic subunit Cna1 and the regulatory subunit Cnb1 [14,15]. Upon exposure to environmental stress, calcineurin mediates the dephosphorylation and subsequent activation of the transcription factor Crz1, thereby regulating the expression of multiple target genes [14–18]. In *Candida albicans* and *Candida auris*, deletion of the *cnb1* gene does not affect biofilm formation *in vitro* [19,20]. On the other hand, biofilm formation by *Aspergillus niger in vitro* requires the calcineurin pathway [21]. These findings indicate that the requirement for calcineurin in biofilm formation differs among fungal species.

In *T. asahii*, calcineurin is essential for growth at 40°C; tolerance to membrane, cell wall, and endoplasmic reticulum stress; and virulence in a silkworm infection model [22]. Morphologic changes of *T. asahii* are also regulated by calcineurin [22]. Tacrolimus, a representative calcineurin inhibitor, suppresses calcineurin signaling in *T. asahii*, leading to high temperature sensitivity, cell wall stress sensitivity, and reduced hyphal formation [23]. The role of calcineurin in biofilm formation by *T. asahii* based on genetic approaches using gene-deficient mutants, however, has not yet been elucidated.

In the present study, we found that while the mutants deficient for *cna1* and *cnb1* genes formed biofilms *in vitro*, *in vivo* biofilm formation was reduced in the catheter-inserted silkworm infection model. Similarly, tacrolimus did not affect biofilm formation by *T. asahii in vitro* but inhibited biofilm formation *in vivo* in the catheter-inserted silkworm infection model. Together, these findings suggest that calcineurin is essential for biofilm formation by *T. asahii* under host environmental conditions.

## Materials and methods

### Reagents

Cefotaxime sodium, D-glucose, agar, crystal violet, acetic acid, and tacrolimus (FK506) were purchased from Fujifilm Wako Pure Chemical Industries (Osaka, Japan). G418 was purchased from Enzo Life Science, Inc. (Farmingdale, NY, USA).

Hipolypeptone was purchased from Nihon Pharmaceutical Co., Ltd. (Tokyo, Japan). Calcofluor White (CFW) stain solution was purchased from Sigma-Aldrich (St. Louis, MO, USA). Tacrolimus powder was suspended in saline.

### *T. asahii* strains and culture condition

The *T. asahii* strains used in this study were generated as previously described [22]. Information on these strains is provided in Table 1. The *ku70* gene-deficient strain was used as the parental strain in this study. The *ku70*-deficient strain serves as a suitable parental strain for genetic manipulation and facilitates genetic analyses of *T. asahii* [24,25]. The *cna1* or *cnb1* gene-deficient strains were generated by replacing each target gene using 5'-UTR (*cna1*) -NAT1–3'-UTR (*cna1*) or 5'-UTR (*cnb1*) -NAT1–3'-UTR (*cnb1*) fragments [22]. Complemented strains were generated by reintroducing each target gene using 5'-UTR (*cna1*) -cna1-hph-3'-UTR (*cna1*) or 5'-UTR (*cnb1*) -cnb1-hph-3'-UTR (*cnb1*) fragments [22]. Culture of *T. asahii* strains was performed according to the previously reported method [22]. The strains were grown on Sabouraud dextrose agar (SDA; 1% hipolypeptone, 4% dextrose, and 1.5% agar) containing G418 (100 µg/mL) and incubated at 27°C for 2 days.

### Adhesion assay

The *T. asahii* strains were grown on SDA at 27°C for 2 days. Cells were suspended in physiologic saline and passed through a 40-µm cell strainer (Corning Inc., Corning, NY, USA). The cell suspension was adjusted to an absorbance of 0.25 at 630 nm using Sabouraud medium. Filtered cell suspensions (100 µL) were added to wells of a 96-well microtiter plate (Techno Plastic Products, Trasadingen, Switzerland) and incubated at 37°C for 1 h. After incubation, the supernatants were removed and the wells were washed with phosphate-buffered saline (PBS). Adhesion was quantified using crystal violet staining. A 0.1% crystal violet solution (50 µL) was added to the dried wells and incubated for 60 min. The wells were then washed three times with PBS and air-dried for 30 min. Acetic acid (33%, 100 µL) was added to solubilize the dye, and absorbance at 550 nm was measured using a microplate reader (iMark; Bio-Rad Laboratories Inc., Hercules, CA, USA).

### Biofilm measurements *in vitro*

The biofilm formation assay was performed as described previously [1]. The *T. asahii* strains were grown on SDA at 27°C for 2 days. The *T. asahii* cells were suspended in physiologic saline solution and filtered through a 40-µm cell strainer (Corning Inc.). The cell suspension was adjusted to an absorbance of 0.1 at 630 nm with Sabouraud medium. The filtered cell suspensions (100 µL) were applied to wells of 96-well microtiter plate (Techno Plastic Products) and incubated at 37°C for 1 h. After incubation, the supernatants were removed. The wells containing *T. asahii* cells were washed with PBS and fresh medium was added. After incubation at 37°C for 24 h, the supernatant was removed and replaced with fresh medium. After another 24-h incubation, planktonic cells were removed, and the wells were washed three times with PBS. Biofilm mass was measured using crystal violet. The 0.1% crystal violet solution (50 µL) was added to the dried wells, and the plate was incubated for 30 min. After incubation, the wells were washed three times with PBS and dried for 30 min.

**Table 1.  *T. asahii* strains used in this study.**

| *T. asahii* strains | Genotype | Reference |
|---|---|---|
| MPU129 Δ*ku70* | *ku70*::nptII | [24] |
| (Parent strain) | *ku70*::nptII, cna1::NAT1 | [22] |
| Δ*cna1* | *ku70*::nptII, NAT1::cna1, hph | [22] |
| Comp. (CNA1) | *ku70*::nptII, cnb1::NAT1 | [22] |
| Δ*cnb1* | *ku70*::nptII, NAT1::cnb1, hph | [22] |
| Comp. (CNB1) | | |

Acetic acid solution (33%, 50 µL) was added to the wells and absorbance at 550 nm was measured using a microplate reader (iMark microplate reader).

## Observation of *T. asahii* morphology in biofilms

The morphology of *T. asahii* in the biofilms was observed as described previously [12]. *T. asahii* cells were suspended in physiologic saline and passed through a 40-µm cell strainer (Corning Inc.). The filtered cell suspension was adjusted to an absorbance of 0.1 at 630 nm using Sabouraud medium. Cell suspensions (100 µL) were added to wells of a 96-well microtiter plate (Techno Plastic Products) and incubated at 37°C for 1 h. After incubation, the supernatants were removed, the wells were washed with PBS, and fresh medium was added. The plates were incubated at 37°C for 24 h, and then the supernatants were replaced with fresh medium and incubation was continued for an additional 24 h. Following incubation, planktonic cells were removed, and the wells were washed three times with PBS. Calcofluor White stain solution (100 µL) was added to the wells, and the plates were incubated at 25°C for 15 min. Biofilms were observed and photographed using a fluorescence microscope (BZ-X800; Keyence Corporation, Osaka, Japan). Fluorescence intensity was analyzed using ImageJ software (version 1.47t; National Institutes of Health, Bethesda, MD, USA).

## *In vivo* biofilm formation by *T. asahii* using the silkworm infection model

The *in vivo* biofilm formation assay was performed as previously described [12]. Eggs of silkworms (KINSYU×SHOWA) were purchased from Ehime-Sanshu Co., Ltd. (Ehime, Japan). Fifth instar larvae were fed overnight with an artificial diet (Silkmate 2S; Ehime-Sanshu Co., Ltd.). A polyurethane fiber (0.5 mm thick, Gomutegusu F046, No. H3; Daiso-Sangyo, Hiroshima, Japan) was cut into 2-cm segments, treated with 70% ethanol for 15 min, and dried under UV irradiation for 30 min. A small hole was made on the dorsal surface of each silkworm using a marking pin (Daiso-Sangyo), and a UV-sterilized polyurethane fiber was inserted under the cuticle into the hemolymph. The fiber-inserted silkworms were maintained at 25°C for 30 min to confirm cessation of bleeding. *T. asahii* cells grown on SDA plates for 1 day at 27°C were suspended in physiologic saline and filtered through a 40-µm cell strainer (Corning Inc.). Silkworms were injected with 50 µL of the cell suspension ($A_{630}$ = 1) and incubated at 27°C for 24 h. The polyurethane fibers were recovered, transferred to 1.5-mL tubes, washed twice with saline, and fixed with methanol for 20 min. After methanol removal, fibers were air-dried for 1 h. A 0.1% (w/v) crystal violet solution (350 µL) was added and the fibers were incubated at 25°C for 20 min. After removing the staining solution, fibers were washed twice with 20% ethanol and once with distilled water. Biofilms formed on fiber surfaces were observed using a light microscope (CH-30; Olympus, Tokyo, Japan). After microscopic observation, fibers were placed in 33% (v/v) acetic acid (500 µL) for 30 min, mixed with distilled water (500 µL), and absorbance at 590 nm was measured.

For tacrolimus administration test, *T. asahii* cells grown on SDA plates for 1 day at 27°C were suspended in physiologic saline and filtered through a 40-µm cell strainer (Corning Inc.). Silkworms were injected with 50 µL of the cell suspension ($A_{630}$ = 1). After injection of the *T. asahii* cells, saline or tacrolimus solution (50 µg/50 µL) were administered, and the silkworms were incubated at 27°C for 24 h. After incubation, PFs were isolated from the silkworms, stained with crystal violet.

## Statistical analysis

All experiments were performed at least three times, and representative results are presented. Statistical differences among multiple groups (see Figs 1A, 1C, 2B, 3C and 4B) were evaluated using Tukey's test. Differences shown in Fig 5B were assessed using Student's *t*-test. Statistical significance was defined as $P < 0.05$.

## Results

### Effect of calcineurin deficiency on *T. asahii* biofilm formation *in vitro*

In *Candida albicans*, calcineurin is not required for biofilm formation *in vitro* [19]. Therefore, we examined whether deletion of the *cna1* and the *cnb1* gene affects biofilm formation by *T. asahii in vitro*. The *cna1* and the *cnb1* gene-deficient mutant

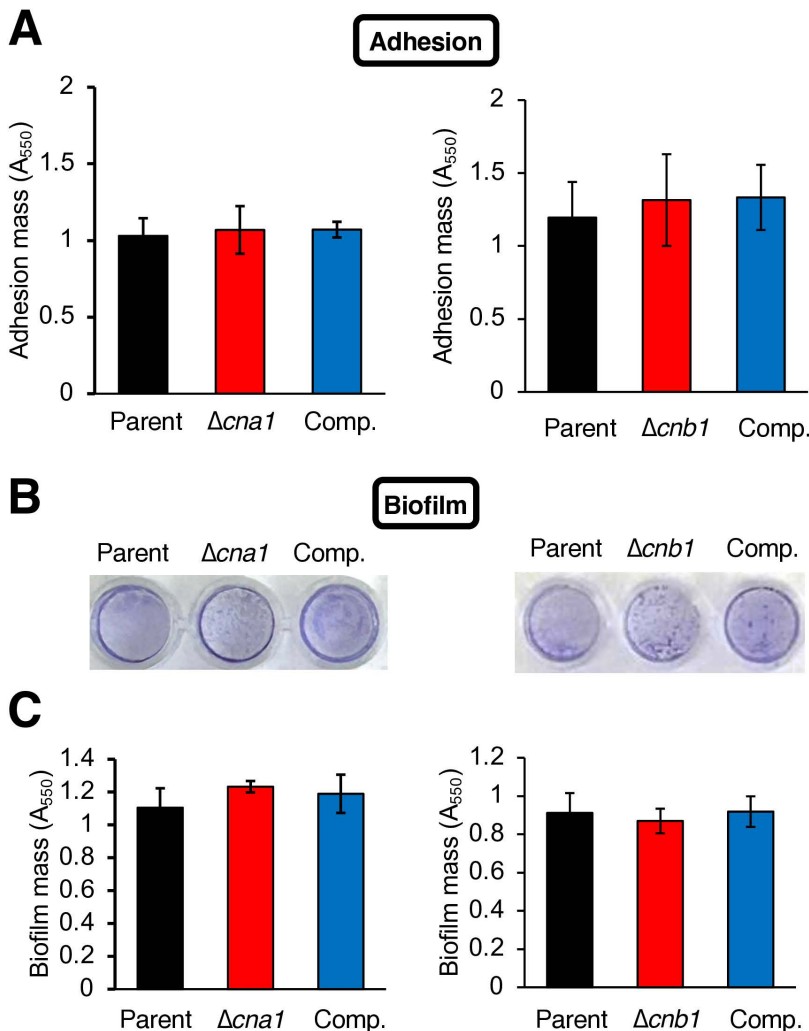

**Fig 1. Adhesion and biofilm formation in the *T. asahii cna1* and *cnb1* gene-deficient mutants *in vitro*. (A)** *T. asahii* cell suspensions were adjusted to an absorbance of 0.5 at 630 nm using Sabouraud medium. Aliquots (100 μL) were added to wells of a 96-well microtiter plate and incubated at 37°C for 1 h. After incubation, non-adherent cells were removed, and adhesion was quantified using crystal violet (CV) staining. The amount of CV retained was determined by measuring absorbance at 550 nm ($A_{550}$). $n = 3$/group. **(B, C)** Biofilm formation by the parent strain (Parent), the *cna1* gene-deficient mutant (Δ*cna1*), the *cnb1* gene-deficient mutant (Δ*cnb1*), and their respective complemented strains (Comp.) in Sabouraud dextrose medium was assessed based on CV staining. **(B)** Representative images of CV-stained biofilms are shown. **(C)** CV retention was quantified by measuring absorbance at 550 nm ($A_{550}$). Data are presented as the mean ± standard deviation (SD). Statistical significance was assessed using the Tukey's test. *: $P < 0.05$. $n = 3$/group.

showed comparable adhesion to polyethylene in Sabouraud dextrose medium relative to the parent strain (Fig 1A). In addition, no differences in biofilm biomass were observed between the mutant and parent strains (Fig 1B and 1C). Biofilms are composed of fungal cells and an extracellular matrix consisting of polysaccharides, proteins, nucleic acids, and lipids [26,27]. To evaluate fungal cell mass within the biofilm, we quantified cell-associated chitin using Calcofluor White staining. Fluorescence intensity of the mutants deficient for the *cna1* and *cnb1* genes were not decreased compared with that of the parent strain (Fig 2). These findings suggest that Cna1 and Cnb1 do not regulate biofilm formation by *T. asahii in vitro*.

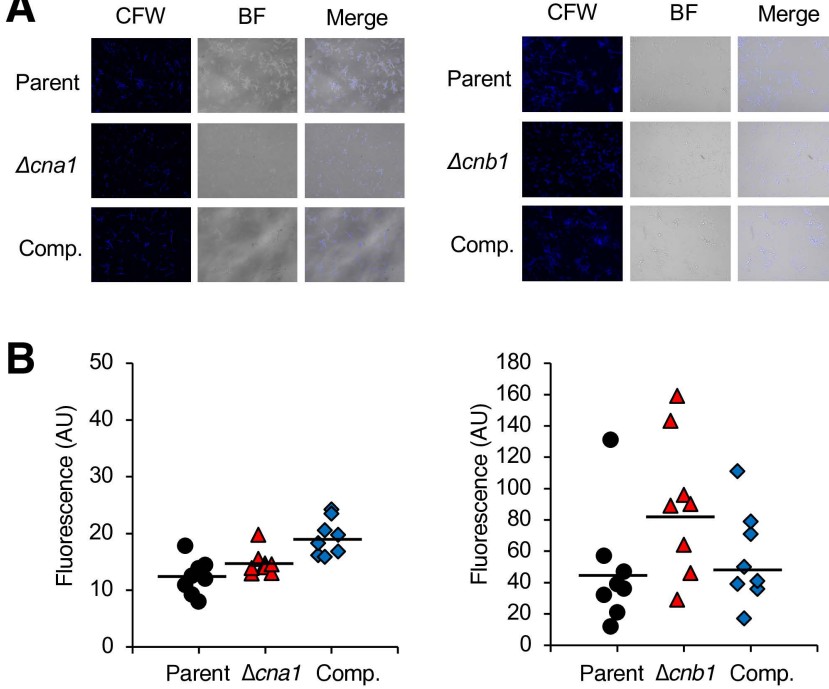

**Fig 2. Morphology of the *cnb1* gene-deficient mutant in biofilms.** *T. asahii* cells stained with Calcofluor White (CFW) were observed using fluorescence microscopy. Biofilms formed by the parent strain (Parent), the *cna1* gene-deficient mutant (Δ*cna1*), the *cnb1* gene-deficient mutant (Δ*cnb1*), and their respective complemented strains (Comp.) were examined. **(A)** Representative fluorescence microscopy images obtained at 40×magnification. **(B)** Quantification of CFW fluorescence intensity. Statistical significance was assessed using the Tukey's test. *: $P < 0.05$. $n = 8$/group.

### Essential role of the calcineurin in biofilm formation by *T. asahii* in the *in vivo* silkworm model

During infection, *T. asahii* is exposed to a variety of host-derived stressors, including oxidative stress [28]. Using the catheter-inserted silkworm infection model, we assessed whether deletion of the *cna1* gene affects biofilm formation *in vivo*. Biofilm formation by the *cna1* gene-deficient mutant was markedly reduced compared with that by the parent strain (Fig 3). The phenotype of the complemented strain was comparable to that of the parent strain (Fig 3). Similar results were observed for the *cnb1* gene-deficient mutant (Fig 3). These findings indicate that calcineurin is required for biofilm formation by *T. asahii in vivo*.

### Effects of tacrolimus on biofilm formation by *T. asahii in vitro*

We next examined the effects of tacrolimus, a calcineurin inhibitor, on biofilm formation by *T. asahii in vitro*. Tacrolimus inhibits calcineurin by binding to FKBP12 [23]. The *cnb1* gene-deficient mutant shows a growth delay at 40°C [22]. Under this condition, addition of tacrolimus (0.16–10 μg/mL) to Sabouraud agar delayed the growth of the parent strain but not that of the *cnb1* gene-deficient mutant (Fig 4A). On the other hand, tacrolimus (0.08–10 μg/mL) did not inhibit biofilm formation by the parent strain *in vitro* (Fig 4B). These findings indicate that tacrolimus does not suppress biofilm formation by *T. asahii* under *in vitro* culture conditions.

### Inhibition of *T. asahii* biofilm formation by tacrolimus in an *in vivo* silkworm model

We next examined whether tacrolimus inhibits biofilm formation by *T. asahii in vivo*. Biofilm formation by the parent strain on the surface of the polyurethane fiber in silkworm hemolymph was reduced by the administration of tacrolimus (Fig 5).

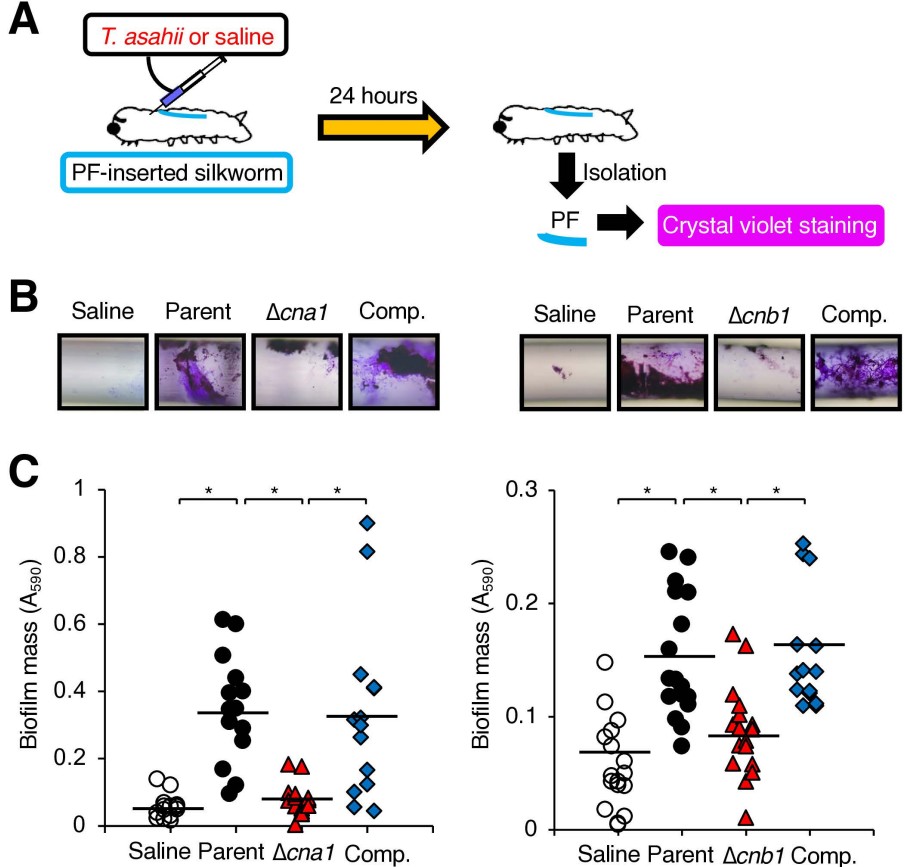

**Fig 3. Effects of *cna1* or *cnb1* gene deficiency on *T. asahii* biofilm formation *in vivo*.** **(A)** Schematic representation of the *in vivo* biofilm assay using silkworms. Polyurethane fiber (PF)-inserted silkworms were prepared, and cell suspensions ($A_{630} = 1$, 50 µL) of the parent strain (Parent), the *cna1* gene-deficient mutant (Δ*cna1*), *cnb1* gene-deficient mutant (Δ*cnb1*), and their respective complemented strains (Comp.) were injected. PF-inserted silkworms were then incubated at 27°C for 24 h. **(B)** PFs recovered from silkworms were stained with crystal violet and observed under a microscope. **(C)** Absorbance of the eluted dye was measured at 590 nm. Statistical significance was assessed using the Tukey's test. *: $P < 0.05$. $n = 14$ or 17/group.

The complemented strain showed a phenotype similar to that of the parent strain (Fig 5). These results suggest that tacrolimus inhibits biofilm formation by *T. asahii in vivo*.

## Discussion

In this study, we investigated the role of calcineurin in biofilm formation by *T. asahii in vitro* and *in vivo*. In the *in vitro* assay, the *cna1* and the *cnb1* genes was not required for biofilm formation. In contrast, in the *in vivo* silkworm model, the *cna1* and the *cnb1* gene were required for biofilm formation on the surface of a polyurethane fiber. Furthermore, tacrolimus, a calcineurin inhibitor, did not inhibit biofilm formation *in vitro*, but significantly inhibited biofilm formation *in vivo*. These findings suggest that calcineurin plays a critical role in biofilm formation by *T. asahii* under host environmental conditions and that tacrolimus-mediated calcineurin inhibition impairs biofilm formation *in vivo*.

Calcineurin influences biofilm formation by *T. asahii in vivo*, but not *in vitro*. In the *in vitro* biofilm formation abilities, no significant difference in biofilm formation on the surface of polyurethane fibers was observed between the parent strain and the calcineurin-deficient mutants (Fig. S1 in S1 File). The *cna1* and *cnb1* gene-deficient mutants exhibited sensitivity to membrane damaging agent sodium dodecyl sulfate (SDS), cell wall stress induced by Congo red (CR), oxidative stress

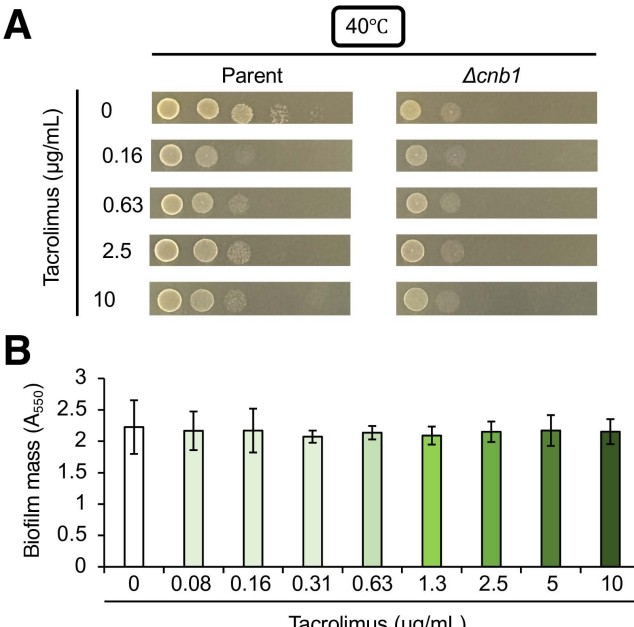

**Fig 4. Effect of tacrolimus on the *T. asahii* growth and biofilm formation *in vitro*. (A)** The *T. asahii* parent strain (Parent) and the *cnb1* gene-deficient mutant (Δ*cnb1*) were grown on SDA and incubated at 27°C for 1 day. *T. asahii* cells were suspended in physiologic saline solution and filtered through a 40-μm cell strainer. A series of 10-fold dilutions of the fungal suspension were prepared in saline. Five microliters of each cell suspension were spotted on the SDA containing tacrolimus (0-10 mg/mL). Agar plates were incubated at 40°C for 24 **h. (B)** Biofilm formation by *T. asahii* in Sabouraud dextrose medium *in vitro* was determined by crystal violet (CV) staining. The amounts of biofilm formed by the parent strain in Sabouraud dextrose medium containing tacrolimus (0-10 mg/mL) were determined by CV staining. The CV was quantified by measuring the absorbance at 550 nm ($A_{550}$). Data are shown as means ± standard deviation (SD). Statistical significance was assessed using the Tukey's test. *: $P < 0.05$. n = 5/group.

mediated by hydrogen peroxide ($H_2O_2$), and endoplasmic reticulum stress caused by tunicamycin and dithiothreitol [22]. In addition, virulence of *T. asahii* in the silkworm infection model is reduced in the *cna1* gene- and the *cnb1* gene-deficient mutants [22]. These observations suggest that calcineurin is required for adaptation of *T. asahii* to host-associated stress conditions. Because calcineurin contributes to tolerance against stress-inducing compounds such as $H_2O_2$, dithiothreitol, tunicamycin, Congo red, and SDS, these stressors may activate the calcineurin signaling pathway. However, the addition of these compounds did not significantly enhance biofilm formation by *T. asahii* (Fig. S2 in S1 File). In *Neurospora crassa*, Cna1 is essential for female reproduction, whereas Cnb1 is dispensable for this process [29]. Based on this functional divergence, we examined both *cna1* and *cnb1* gene-deficient mutants in the present study. Under the experimental conditions tested, we did not observe distinct roles for the *cna1* and *cnb1* genes in biofilm formation by *T. asahii*. These findings suggest that calcineurin, as a functional complex, regulates in vivo biofilm formation in *T. asahii* under the conditions examined in this study. The type strain JCM2466 exhibits more than a 10-fold lower virulence in the silkworm infection model than the MPU129 strain used in this study [24]. The JCM2466 did not show reduced biofilm formation in silkworms compared to the MPU129 strain (Fig. S3 in S1 File). This result suggests that reduced virulence does not necessarily correlate with decreased biofilm formation in the silkworm model. The precise mechanisms by which calcineurin contributes to biofilm formation by *T. asahii in vivo* remain to be elucidated and will be an important subject of future research.

*In vitro*, where *T. asahii* is cultured in nutrient-rich medium such as Sabouraud medium and is not exposed to host-related stresses, biofilm formation can occur independently of calcineurin. In contrast, in the *in vivo* silkworm model, biofilm formation likely depends on calcineurin-mediated adaptation to host environmental stresses. Similar findings have been reported for *Candida albicans*, in which calcineurin deficiency does not affect biofilm formation *in vitro* but impairs

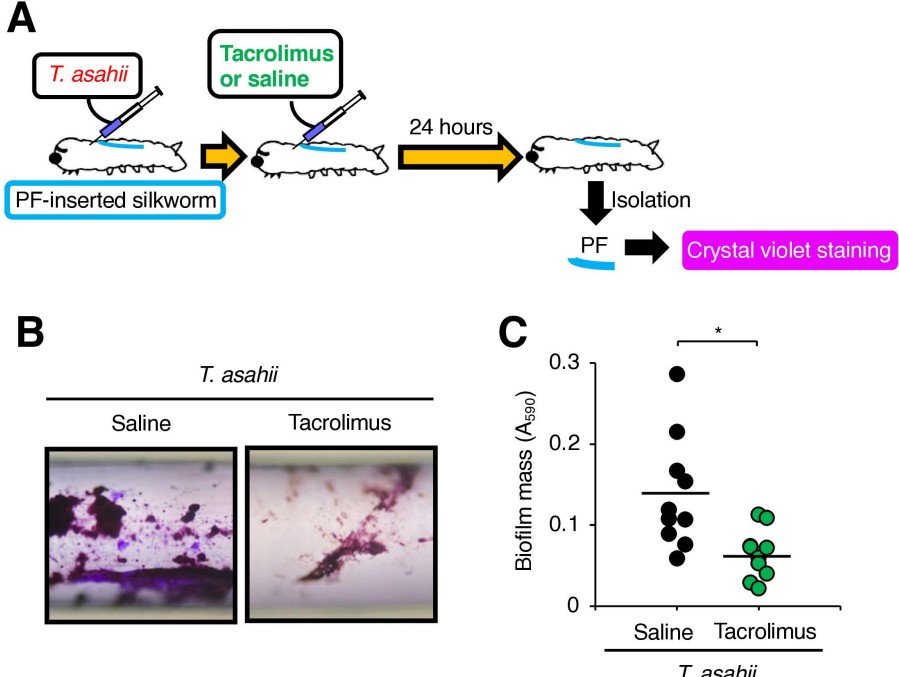

**Fig 5. Effect of tacrolimus on *T. asahii* biofilm formation *in vivo*.** (A) Experimental scheme of the *in vivo* biofilm assay using silkworms. Polyurethane fiber (PF)-inserted silkworms were prepared. Cell suspensions ($A_{630}$ = 1) (50 μL) of the parent strain (Parent) were injected into the PF-inserted silkworms. After injection of the *T. asahii* cells, saline or tacrolimus solution (50 μg/50 μL) were administered, and the silkworms were incubated at 27°C for 24 h. After incubation, PFs were isolated from the silkworms, stained with crystal violet, and observed under a microscope (B). The absorbance of the eluted dye was measured at 590 nm (C). Statistically significant differences between groups were evaluated using Student's *t*-test. *: $P < 0.05$. n = 10/ group.

biofilm formation in an *in vivo* catheter-inserted rat model [19]. Therefore, the finding that *in vivo* biofilm formation in *T. asahii* is calcineurin-dependent is consistent with findings in *C. albicans*. These observations suggest that calcineurin may serve as a potential therapeutic target for preventing *in vivo* biofilm formation by both *T. asahii* and *C. albicans*. *Cryptococcus neoformans*, which belongs to the same phylum Basidiomycota as *Trichosporon asahii*, is also capable of forming biofilms [30]. Moreover, calcineurin is involved in various stress tolerance and virulence in *C. neoformans* [16]. These observations and our results raise the possibility that calcineurin may contribute to biofilm formation by *C. neoformans* under host environmental conditions.

Tacrolimus inhibited biofilm formation by *T. asahii* in the *in vivo* silkworm model. We speculate that inhibiting calcineurin by administering tacrolimus reduces the ability of *T. asahii* to adapt to host environmental stresses, thereby impairing biofilm formation *in vivo* (Fig 6). Tacrolimus treatment altered colony morphology to resemble that of the *cnb1* gene-deficient mutant but did not significantly reduce growth at 27°C (Fig. S4 in S1 File). These findings suggest that fungal calcineurin-targeting compounds may have potential for inhibiting biofilm formation by *T. asahii in vivo*. Because tacrolimus exerts immunosuppressive toxicity in humans by suppressing T-cell activation [31], a low-immunosuppressive FK506 analog, APX879, which inhibits *Aspergillus fumigatus* calcineurin, has been developed based on comparative structural analyses [32,33]. Structure-guided drug design approaches may thus enable the development of fungal-specific calcineurin inhibitors. Such fungal-specific inhibitors may also be effective against biofilm formation by *T. asahii in vivo*. However, silkworms are invertebrates and lack T cells. Therefore, we assume that evaluation in mammalian models, such as mice, rather than in silkworms, would allow more appropriate assessment of fungus-specific antifungal compounds with

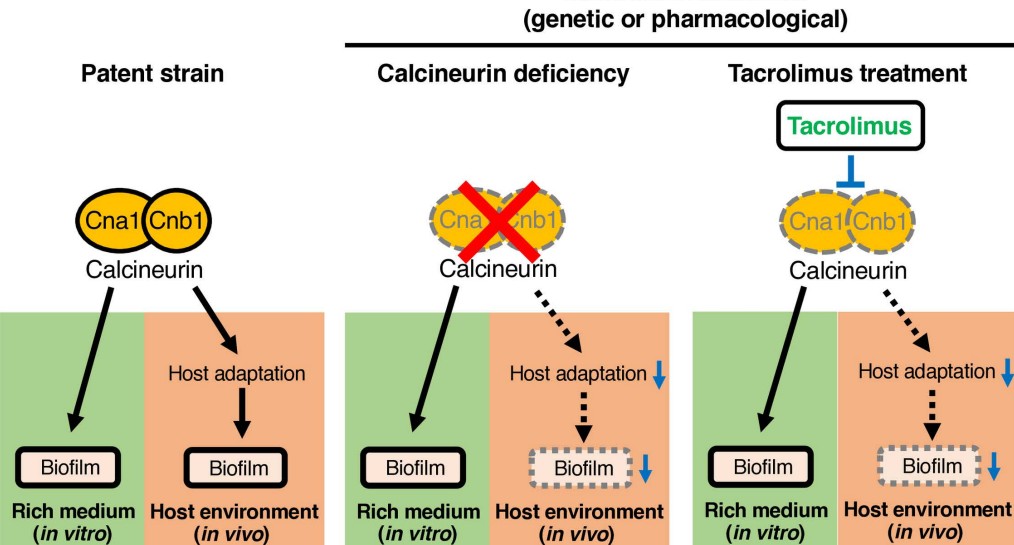

**Fig 6. Genetic and pharmacologic inhibition of calcineurin reduces *T. asahii* biofilm formation.** Tacrolimus inhibits calcineurin in *T. asahii*, which does not affect biofilm formation under nutrient-rich *in vitro* conditions but suppresses biofilm formation *in vivo* under host-derived conditions.

reduced immunosuppressive activity. The development of fungal-specific calcineurin inhibitors will be an important focus of future research.

Cyclosporin A inhibits both calcineurin-dependent and calcineurin-independent pathways by binding to cyclophilin, a fungal peptidylprolyl cis-trans isomerase (PPIase) [34,35]. The cyclosporin A–cyclophilin complex binds to Cna1 and inhibits calcineurin phosphatase activity [34]. Moreover, cyclosporin A impairs mitochondrial function and the permeation transition pore in a calcineurin-independent manner by inhibiting the PPIase activity of cyclophilin [35]. Cyclosporin A inhibits biofilm formation by *T. asahii in vitro* [36]. In the present study, *in vitro* biofilm formation by the *cnb1* gene-deficient mutant was not reduced, suggesting that cyclosporin A may inhibit biofilm formation by *T. asahii in vitro* through a calcineurin-independent mechanism.

## Conclusion

In conclusion, calcineurin plays a critical role in biofilm formation by *T. asahii in vivo*. Calcineurin-deficient mutants exhibited reduced biofilm formation in an *in vivo* silkworm model but did not show reduced biofilm formation under *in vitro* nutrient-rich conditions. Consistently, tacrolimus treatment suppressed biofilm formation *in vivo* but had no inhibitory effect *in vitro*. These findings suggest that calcineurin contributes to the adaptation of *T. asahii* to host environmental conditions required for biofilm formation. Fungal-specific calcineurin inhibitors may be promising candidates for anti-biofilm drug development.

## Supporting information

**S1 File. The file includes Figs. S1-S4.**
(PDF)

**S1 Data. Datasets included in this study.**
(XLSX)

## Acknowledgments

We thank Renta Endo, Momoka Yagi, and Haruka Ogino (Meiji Pharmaceutical University) for technical assistance in rearing the silkworms. We also thank SciTechEdit International LLC (Highlands Ranch, CO, USA) for English language editing.

## Author contributions

**Conceptualization:** Yasuhiko Matsumoto.

**Data curation:** Yasuhiko Matsumoto, Yuta Shimizu, Mei Nakayama, Mai Takizawa.

**Formal analysis:** Yasuhiko Matsumoto, Yuta Shimizu, Mei Nakayama, Mai Takizawa.

**Funding acquisition:** Yasuhiko Matsumoto.

**Investigation:** Yasuhiko Matsumoto, Yuta Shimizu, Mei Nakayama, Mai Takizawa.

**Methodology:** Yasuhiko Matsumoto, Yuta Shimizu, Mei Nakayama, Sanae Kurakado.

**Project administration:** Yasuhiko Matsumoto, Sanae Kurakado.

**Supervision:** Yasuhiko Matsumoto, Takashi Sugita.

**Writing – original draft:** Yasuhiko Matsumoto.

**Writing – review & editing:** Yuta Shimizu, Mei Nakayama, Mai Takizawa, Sanae Kurakado, Takashi Sugita.

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
