## [Decision Letter · Decision Letter 0]

11 Dec 2025

Dear Dr. Matsumoto,

Thank you for submitting your manuscript to PLOS ONE. After careful consideration, we feel that it has merit but does not fully meet PLOS ONE’s publication criteria as it currently stands. Therefore, we invite you to submit a revised version of the manuscript that addresses the points raised during the review process.



We look forward to receiving your revised manuscript.

Kind regards,

Katherine A. Borkovich, Ph.D.

Academic Editor

PLOS One

Journal Requirements:

3.We noticed you have some minor occurrence of overlapping text with the following previous publication(s), which needs to be addressed:

https://pubmed.ncbi.nlm.nih.gov/39567638/

In your revision ensure you cite all your sources (including your own works), and quote or rephrase any duplicated text outside the methods section. Further consideration is dependent on these concerns being addressed.

“This study was supported by JSPS KAKENHI grant number JP23K06141 (Scientific Research (C) to Y.M.) and in part by the Research Program on Emerging and Re-emerging Infectious Diseases of the Japan Agency for Medical Research and Development, AMED (Grant number JP24fk0108679h0402 to T.S.).”

“This study was supported by JSPS KAKENHI grant number JP23K06141 (Scientific Research (C) to Y.M.) and in part by the Research Program on Emerging and Re-emerging Infectious Diseases of the Japan Agency for Medical Research and Development, AMED (Grant number JP24fk0108679h0402 to T.S.).”

6. We note that there is identifying data in the Supporting Information file <file name>. Due to the inclusion of these potentially identifying data, we have removed this file from your file inventory. Prior to sharing human research participant data, authors should consult with an ethics committee to ensure data are shared in accordance with participant consent and all applicable local laws.

-Location data

Reviewers' comments:

Reviewer's Responses to Questions

**Comments to the Author**

1. Is the manuscript technically sound, and do the data support the conclusions?

Reviewer #1: No

Reviewer #2: Partly

2. Has the statistical analysis been performed appropriately and rigorously?

Reviewer #1: Yes

Reviewer #2: Yes

3. Have the authors made all data underlying the findings in their manuscript fully available?

Reviewer #1: Yes

Reviewer #2: No

4. Is the manuscript presented in an intelligible fashion and written in standard English?

Reviewer #1: Yes

Reviewer #2: Yes

Reviewer #1: This manuscript reports that calcineurin deficiency in Trichosporon asahii results in poor biofilm formation on polyurethane fibers (simulating catheters) in an insect model of infection. It follows an earlier study showing that calcineurin deficiency in this organism, an interesting pathogen of humans, produces many phenotypes in vitro and in the same insect model, including loss of hyphae formation and over 10x weaker virulence of cna1/cnb1 mutants. The authors also report that biofilm formation in vitro (in microplate dishes) was equivalent in the calcineurin-deficient mutant and in the tacrolimus-treated wild-type T. asahii. Therefore, the two models of biofilm formation were seemingly discordant in the dependence on calcineurin.

The experiments shown were performed well. However, the conclusion that calcineurin promotes biofilm formation in the host animals (only) is not justified because the earlier paper showed that calcineurin-deficient mutants also fail to form hyphae and fail to thrive in that environment. Therefore, the absence of biofilm in the host animals could be secondary to the absence of hyphae and viability of the calcineurin mutants. In other words, the biofilm defect is expected because calcineurin-deficient T. asahii simply does not thrive in the host environment. In its present form, the manuscript does not represent a significant advance in the field.

Additional concerns

1. Tacrolimus could be injected into host animals after the infection of wild-type T. asahii has already been established to help disentangle the role of calcineurin in different phenotypes (biofilm formation, hyphae formation, virulence). There is speculation that fungal-specific inhibitors of calcineurin would be useful antifungals, so why not test this idea here?

2. Figures 2 and 4 – Genetic and chemical disruption of calcineurin had no effect on biofilm formation in nutrient rich media in vitro. Calcineurin may not be active in these conditions, raising the question of whether the stimulation of calcineurin signaling can increase biofilm formation in vitro. Do stresses (H2O2, DTT, TM, etc) or host factors drive biofilm formation in vitro by increasing calcineurin signaling?

3. Biofilms in vitro were quantified on polystyrene dishes while biofilms in vivo were quantified on polyurethane fibers. So, the differences in vivo and in vitro may depend on the type of plastic rather than the culture conditions. This question should be resolved for better comparisons.

4. Figures 3A and 5A "or saline" should be added to both figures to denote when saline injections occurred.

5. Lines 270-272. The paper cited (ref 19. Uppuluri et al., 2008) shows that the cnb1/cnb1 genetic mutant of C. albicans and WT treated with FK605 (tacrolimus) alone can still form biofilms in the rat-catheter model comparable to wild-type. If the authors disagree with the interpretations in Uppuluri et al. it should be clearly stated and explained in the discussion.

6. Trichosporon is a basidiomycete more closely related to Cryptococcus than to Candida. More direct comparisons to calcineurin signaling in Cryptococcus would be helpful.

Reviewer #2: The manuscript titled “Genetic and pharmacologic inhibition of calcineurin reduces biofilm formation by the pathogenic fungus Trichosporon asahil in an in vivo silkworm infection model” by Matsumoto et al. concluded that calcineurin plays a role in biofilm formation in the T. asahii model for silkworm infection. Interestingly, the author found that calcineurin suppressed biofilm formation in vitro, but not in vivo, in the T. asahil. The manuscript contains extensive research work, conclusive results backed by statistical analysis. However, the manuscript finding is still not fully conclusive due to the fact that the results are mostly derived by using mutant for the regulatory subunit Cnb1 only. In addition, there are some major issues that must be addressed before accepting this manuscript.

Major issues:

The calcineurin protein is consisting of a catalytic subunit (Cna1) and a regulatory subunit (Cnb1). Thus, the authors in this manuscript derived the conclusion based on the results using the deletion mutant for the Cnb1 and inhibition of calcineurin using pharmacological drugs. Therefore, I recommend followings for completing this gap.

(1) The authors did not mention about the mutants for the catalytic subunit. Did the author attempt to generate mutants for the Cna1? Or was it lethal? This should be clearly mentioned at least in the discussion section. It may be noted that previous work also suggested that both Cna1 and Cnb1 function differently, at least in the model filamentous fungus Neurospora crassa, in which Cna1, but not Cnb1, is required for female fertility (Tamuli et al. 2016). Thus, Cna1 mutant could result in biofilm formation even in vitro in the T. asahii model for silkworm infection. The author should discuss this in the light of Tamuli et al. 2016 (PMID: 27019426) or similar references and cite relevant references.

(2) The MPU129 �ku70 was used as a parental strain. However, this contains deletion for the ku70 gene. Could the author justify why the original wild type was not used as a control strain?

Minor comments:

(1) Biofilm formation assay is based on either Calcofluor White (CFW) or absorbance at 590 nm. Did author also perform PCR based assay to confirm biofilm?

(2) Table 1 is confusing. The author should make ideally three columns; Strain, Genotype, and Reference (see below for an example).

Strain Genotype Reference

(3) Figure 1(C): “.......standard deviation (SD), n=3/group” , please specify P value (if applicable) (e. g. P < 0.05).

(4) What was the solvent used to make tarcolimus solutions?

(5) Did tacrolimus affect the normal growth of and development of T. asahii? If yes, please discuss it in the manuscript.

(6) Conclusion is too short. Please few more lines to summarize key findings.

**Do you want your identity to be public for this peer review?** For information about this choice, including consent withdrawal, please see our Privacy Policy

Reviewer #1: No

Reviewer #2: **Yes:** Ranjan Tamuli

---

## [Author Response · Author response to Decision Letter 1]

24 Jan 2026

Response to the comments:

Comments:

Reviewer #1: This manuscript reports that calcineurin deficiency in Trichosporon asahii results in poor biofilm formation on polyurethane fibers (simulating catheters) in an insect model of infection. It follows an earlier study showing that calcineurin deficiency in this organism, an interesting pathogen of humans, produces many phenotypes in vitro and in the same insect model, including loss of hyphae formation and over 10x weaker virulence of cna1/cnb1 mutants. The authors also report that biofilm formation in vitro (in microplate dishes) was equivalent in the calcineurin-deficient mutant and in the tacrolimus-treated wild-type T. asahii. Therefore, the two models of biofilm formation were seemingly discordant in the dependence on calcineurin.

The experiments shown were performed well. However, the conclusion that calcineurin promotes biofilm formation in the host animals (only) is not justified because the earlier paper showed that calcineurin-deficient mutants also fail to form hyphae and fail to thrive in that environment. Therefore, the absence of biofilm in the host animals could be secondary to the absence of hyphae and viability of the calcineurin mutants. In other words, the biofilm defect is expected because calcineurin-deficient T. asahii simply does not thrive in the host environment. In its present form, the manuscript does not represent a significant advance in the field.

Answer: The type strain JCM2466, which exhibits more than a 10-fold lower virulence in the silkworm infection model than the MPU129 strain used in this study, did not show reduced biofilm formation in silkworms (Fig. S3). This result suggests that reduced virulence does not necessarily correlate with decreased biofilm formation in the silkworm model. Therefore, the reduced virulence observed in the calcineurin-deficient mutants does not, by itself, predict decreased biofilm formation in silkworms.

We agree with the reviewer that, in the absence of a clear explanation of the relationship between virulence and biofilm formation in silkworms, the significance of this study may be difficult to understand. To address this concern, we have added an explanation clarifying the relationship between reduced virulence and biofilm formation in silkworms based on these findings in the Discussion section of the revised manuscript (Page 18, lines 291–297).

[Page 18, lines 291–297]

The type strain JCM2466 exhibits more than a 10-fold lower virulence in the silkworm infection model than the MPU129 strain used in this study [24]. The JCM2466 did not show reduced biofilm formation in silkworms compared to the MPU129 strain (Fig. S3). This result suggests that reduced virulence does not necessarily correlate with decreased biofilm formation in the silkworm model. The precise mechanisms by which calcineurin contributes to biofilm formation by T. asahii in vivo remain to be elucidated and will be an important subject of future research.

Additional concerns

1. Tacrolimus could be injected into host animals after the infection of wild-type T. asahii has already been established to help disentangle the role of calcineurin in different phenotypes (biofilm formation, hyphae formation, virulence). There is speculation that fungal-specific inhibitors of calcineurin would be useful antifungals, so why not test this idea here?

Answer: Tacrolimus exerts immunosuppressive effects by inhibiting calcineurin in mammalian T cells, thereby reducing host immune responses. APX879, a tacrolimus derivative, retains antifungal activity while exhibiting reduced immunosuppressive effects on T cells (Juvvadi PR, et al. Nat Commun. 2019). However, silkworms are invertebrates and lack T cells. Therefore, we assume that evaluation in mammalian models, such as mice, rather than in silkworms, would allow more appropriate assessment of fungus-specific antifungal compounds with reduced immunosuppressive activity. We added the sentences in the Discussion section of the revised manuscript (Page 19, lines 318-page 19, lines 327).

[Page 19, lines 318-page 19, lines 327]

Because tacrolimus exerts immunosuppressive toxicity in humans by suppressing T-cell activation [32], a low-immunosuppressive FK506 analog, APX879, which inhibits Aspergillus fumigatus calcineurin, has been developed based on comparative structural analyses [33,34]. Structure-guided drug design approaches may thus enable the development of fungal-specific calcineurin inhibitors. Such fungal-specific inhibitors may also be effective against biofilm formation by T. asahii in vivo. However, silkworms are invertebrates and lack T cells. Therefore, we assume that evaluation in mammalian models, such as mice, rather than in silkworms, would allow more appropriate assessment of fungus-specific antifungal compounds with reduced immunosuppressive activity. The development of fungal-specific calcineurin inhibitors will be an important focus of future research.

2. Figures 2 and 4 – Genetic and chemical disruption of calcineurin had no effect on biofilm formation in nutrient rich media in vitro. Calcineurin may not be active in these conditions, raising the question of whether the stimulation of calcineurin signaling can increase biofilm formation in vitro. Do stresses (H2O2, DTT, TM, etc) or host factors drive biofilm formation in vitro by increasing calcineurin signaling?

Answer: According to the reviewer’s comment, we performed new experiment that T. asahii biofilm formation was enhanced by adding stressors such as H2O2, DTT, TM, and SDS (Fig. S2). The addition of these compounds did not significantly enhance biofilm formation by T. asahii. We added the sentences in the revised manuscript (Page 17, lines 282-page 18, line 285).

[Page 17, lines 282-page 18, line 285]

Because calcineurin contributes to tolerance against stress-inducing compounds such as H₂O₂, dithiothreitol, tunicamycin, Congo red, and SDS, these stressors may activate the calcineurin signaling pathway. However, the addition of these compounds did not significantly enhance biofilm formation by T. asahii (Fig. S2).

3. Biofilms in vitro were quantified on polystyrene dishes while biofilms in vivo were quantified on polyurethane fibers. So, the differences in vivo and in vitro may depend on the type of plastic rather than the culture conditions. This question should be resolved for better comparisons.

Answer: According to the reviewer’s comment, we performed an additional experiment comparing the in vitro biofilm formation abilities of the parent strain and the calcineurin-deficient mutants on the surface of polyurethane fibers (Fig. S1). No significant difference in biofilm formation was observed between the parent strain and the calcineurin-deficient mutants.

According to the reviewer’s comment, we performed new experiment that the in vitro biofilm formation abilities of the parent strain and the calcineurin deficient mutants on the surface of polyurethane fibers were compared (Fig. S1). No significant difference in biofilm formation on the surface of polyurethane fibers was observed between the parent strain and the calcineurin-deficient strain (Page 17, lines 274-277).

[Page 17, lines 275-278]

In the in vitro biofilm formation abilities, no significant difference in biofilm formation on the surface of polyurethane fibers was observed between the parent strain and the calcineurin-deficient mutants (Fig. S1).

4. Figures 3A and 5A "or saline" should be added to both figures to denote when saline injections occurred.

Answer: Following the reviewer’s comment, we added the words “or saline” in Figures 3A and 5A of the revised manuscript.

5. Lines 270-272. The paper cited (ref 19. Uppuluri et al., 2008) shows that the cnb1/cnb1 genetic mutant of C. albicans and WT treated with FK605 (tacrolimus) alone can still form biofilms in the rat-catheter model comparable to wild-type. If the authors disagree with the interpretations in Uppuluri et al. it should be clearly stated and explained in the discussion.

Answer: Based on our experimental results, we do not disagree with the findings reported by Uppuluri et al. The study by Uppuluri et al. differed from our results not only in the fungal species examined but also in the experimental conditions. In particular, Uppuluri et al. investigated Candida albicans biofilm formation within the lumen of catheters (Uppuluri P., et al., Antimicrob Agents Chemother., 2008). In contrast, our study evaluated biofilm formation by Trichosporon asahii on the surface of polyurethane fibers implanted within the silkworm body. Therefore, in our silkworm model, the catheter material was likely more directly exposed to host cells and host-derived proteins.

Without additional data examining whether calcineurin is required for C. albicans biofilm formation on catheter surfaces fully exposed to blood flow in mammalian models, such as rat veins, direct comparisons between the two studies are difficult. Accordingly, we do not consider our findings to be contradictory to those reported by Uppuluri et al.

6. Trichosporon is a basidiomycete more closely related to Cryptococcus than to Candida. More direct comparisons to calcineurin signaling in Cryptococcus would be helpful.

Answer: According to the reviewer’s comment, we added the sentences in the Discussion section of the revised manuscript (Page 19, lines 307-311).

[Page 19, lines 307-311]

Cryptococcus neoformans, which belongs to the same phylum Basidiomycota as Trichosporon asahii, is also capable of forming biofilms [31]. Moreover, calcineurin is involved in various stress tolerance and virulence in C. neoformans [16]. These observations and our results raise the possibility that calcineurin may contribute to biofilm formation by C. neoformans under host environmental conditions.

Reviewer #2: The manuscript titled “Genetic and pharmacologic inhibition of calcineurin reduces biofilm formation by the pathogenic fungus Trichosporon asahil in an in vivo silkworm infection model” by Matsumoto et al. concluded that calcineurin plays a role in biofilm formation in the T. asahii model for silkworm infection. Interestingly, the author found that calcineurin suppressed biofilm formation in vitro, but not in vivo, in the T. asahil. The manuscript contains extensive research work, conclusive results backed by statistical analysis. However, the manuscript finding is still not fully conclusive due to the fact that the results are mostly derived by using mutant for the regulatory subunit Cnb1 only. In addition, there are some major issues that must be addressed before accepting this manuscript.

Major issues:

The calcineurin protein is consisting of a catalytic subunit (Cna1) and a regulatory subunit (Cnb1). Thus, the authors in this manuscript derived the conclusion based on the results using the deletion mutant for the Cnb1 and inhibition of calcineurin using pharmacological drugs. Therefore, I recommend followings for completing this gap.

(1) The authors did not mention about the mutants for the catalytic subunit. Did the author attempt to generate mutants for the Cna1? Or was it lethal? This should be clearly mentioned at least in the discussion section. It may be noted that previous work also suggested that both Cna1 and Cnb1 function differently, at least in the model filamentous fungus Neurospora crassa, in which Cna1, but not Cnb1, is required for female fertility (Tamuli et al. 2016). Thus, Cna1 mutant could result in biofilm formation even in vitro in the T. asahii model for silkworm infection. The author should discuss this in the light of Tamuli et al. 2016 (PMID: 27019426) or similar references and cite relevant references.

Answer: According to the reviewer’s comment, we performed additional experiments using the cna1 gene-deficient mutant (Figs. 1–3). Similar to the cnb1 gene-deficient mutant, the cna1 gene-deficient mutant did not exhibit reduced biofilm formation in vitro but showed significantly reduced biofilm formation in vivo in the silkworm model (Figs. 1–3).

Following the reviewer’s comment, we added the sentences in the Discussion section of the revised manuscript (Page 18, lines 286-291). In Neurospora crassa, Cna1 is essential for female reproduction, whereas Cnb1 is not (Tamuli et al., 2016). Therefore, we included both cna1 and cnb1 gene-deficient mutants in the present study. Under the experimental conditions tested, we did not observe distinct roles for the cna1 and cnb1 genes in biofilm formation by T. asahii. Accordingly, we conclude that calcineurin as a whole regulates in vivo biofilm formation in T. asahii under the conditions examined in this study.

[Page 18, lines 286-291]

In Neurospora crassa, Cna1 is essential for female reproduction, whereas Cnb1 is dispensable for this process [30]. Based on this functional divergence, we examined both cna1 and cnb1 gene-deficient mutants in the present study. Under the experimental conditions tested, we did not observe distinct roles for the cna1 and cnb1 genes in biofilm formation by T. asahii. These findings suggest that calcineurin, as a functional complex, regulates in vivo biofilm formation in T. asahii under the conditions examined in this study.

(2) The MPU129 �ku70 was used as a parental strain. However, this contains deletion for the ku70 gene. Could the author justify why the original wild type was not used as a control strain?

Answer: We have established a method for generating gene-deficient strains of T. asahii (Matsumoto Y., et al., Sci Rep, 2021, Matsumoto Y., et al., AMB Express, 2022). In our previous studies, we demonstrated that deletion of the ku70 gene, which encodes Ku70 and is required for nonhomologous end joining, significantly increases the efficiency of homologous recombination in T. asahii (Matsumoto Y., et al., Sci Rep, 2021, Matsumoto Y., et al., AMB Express, 2022). We also confirmed that the ku70gene-deficient mutant does not affect the growth of T. asahii or its pathogenicity in the silkworm infection model (Matsumoto Y., et al., Sci Rep, 2021). Therefore, the ku70-deficient strain serves as a suitable parental strain for genetic manipulation and facilitates genetic analyses of T. asahii. Accordingly, we used the ku70-deficient strain as the parental strain in this study, as described in the Materials and Methods section (Page 6, lines 89–91).

[Page 6, lines 89–91]

The ku70 gene-deficient strain was used as the parental strain in this study. The ku70-deficient strain serves as a suitable parental strain for genetic manipulation and facilitates genetic analyses of T. asahii [24,25].

Minor comments:

(1) Biofilm formation assay is based on either Calcofluor White (CFW) or absorbance at 590 nm. Did author also perform PCR based assay to confirm biofilm?

Answer: T. asahii biofilms consist of a mixture of yeast and hyphal cells. Although yeast and hyphal cells differ substantially in cell volume, both are single-nucleated cells and therefore contain comparable amounts of genomic DNA. In this study, biofilm formation was evaluated by measuring total biofilm mass, and DNA-based quantification methods, such as PCR based assay, were not performed.

(2) Table 1 is confusing. The author should make ideally three columns; Strain, Genotype, and Reference (see below for an example).

Strain Genotype Reference

Answer: Following the reviewer’s comment, we changed the description in the Table 1 of the revised manuscript.

(3) Figure 1(C): “.......standard deviation (SD), n=3/group” , please specify P value (if applicable) (e. g. P < 0.05).

Answer: Following the reviewer’s comment, we added the P value in the Figure legend of Figure 1C.

(4) What was the solvent used to make tarcolimus solutions?

Answer: Following the reviewer’s comment, we added information regarding the solvent used for tacrolimus to the Materials and Methods section of the revised manuscript (Page 5, line 85). Tacrolimus powder was suspended in saline.

[Page 5, line 85]

Tacrolimus powder was suspended in saline.

(5) Did tacrolimus affect the normal growth of and development of T. asahii? If yes, please discuss it in the manuscript.

Answer: According to the reviewer’s comment, we performed an additional experiment to examine the effects of tacrolimus

---

## [Decision Letter · Decision Letter 1]

8 Feb 2026

Dear Dr. Matsumoto,

Thank you for submitting your manuscript to PLOS ONE. After careful consideration, we feel that it has merit but does not fully meet PLOS ONE’s publication criteria as it currently stands. Therefore, we invite you to submit a revised version of the manuscript that addresses the points raised during the review process.

We look forward to receiving your revised manuscript.

Kind regards,

Katherine A. Borkovich, Ph.D.

Academic Editor

PLOS One

**Journal Requirements:**

Reviewers' comments:

Reviewer's Responses to Questions

**Comments to the Author**

Reviewer #2: All comments have been addressed

2. Is the manuscript technically sound, and do the data support the conclusions?

Reviewer #2: Yes

3. Has the statistical analysis been performed appropriately and rigorously?

Reviewer #2: Yes

4. Have the authors made all data underlying the findings in their manuscript fully available?

Reviewer #2: No

5. Is the manuscript presented in an intelligible fashion and written in standard English?

Reviewer #2: (No Response)

Reviewer #2: The manuscript titled “Genetic and pharmacologic inhibition of calcineurin reduces biofilm formation by the pathogenic fungus Trichosporon asahii in an in vivo silkworm infection model” by Matsumoto has been revised by taking care of the suggestions made in the initial version. The manuscript has been considerably improved. However, I recommend for additional corrections as suggested below. The strains used must be described well in the Materials and Methods section, particularly, the complementation strain generation procedure used for generating the complemented strains. In addition, please check the language thoroughly in the final version.

1. Page 5: Line 72: “In the present study, we found that while the cnb1 gene-deficient T. asahii mutant.” In the last paragraph of the introduction, only the work done on cnb1 has been described. Please also described the work done on cna1, which is now described in the results section of the revised manuscript.

2. Page 6: Lines 87-94: Please change “T. asahii cultures” as “Generation of the T. asahii strains and culture condition”.

In addition, please briefly describe the mutants and how the complement strains were generated. The author cited reference by Matsumoto et al 2022; however, the complement strain generation was not described. Please describe the procedure to generate the complement strain well.

3. Page 11: lines 182-183: “Fluorescence intensity of the cna1 gene- and the cnb1 gene-deficient mutants were not decreased compared...........”, please correct as “Fluorescence intensity of the mutants deficient for the cna1 and cnb1 genes were not decreased compared....”

4. Page 11: lines 187-188 “Fig. 1. Adhesion and biofilm formation by cna1 gene- and cnb1 gene-deficient mutants of T. asahii in vitro.”, please correct as “Fig. 1. Adhesion and biofilm formation in the T. asahii cna1 and cnb1 mutants.”

5. Page 17: Lines 278-279: “The cna1 gene- and cnb1 gene-deficient mutant exhibits sensitivity to several stresses, including membrane damage (sodium dodecyl sulfate), cell wall stress (Congo red), oxidative stress (H₂O₂), and endoplasmic reticulum stress (tunicamycin and dithiothreitol) [22].”, please correct as “The cna1 and cnb1 deficient mutant exhibited sensitivity to membrane damaging agent sodium dodecyl sulfate (SDS), cell wall stress induced by Congo red (CR), oxidative stress mediated by hydrogen peroxide (H₂O₂), and endoplasmic reticulum stress caused by tunicamycin and dithiothreitol [22].”

6. Page 21: Lines 338-340: Figure 6: The model described only the effect of tacrolimus. However, the author used the cna1 and cnb1 mutants, and the results obtained using these mutants constitute the major part of the manuscript. Therefore, I recommend improving the model by incorporating the results using the mutants.

7. In several figures, the cna1, cnb1 are shown within a box on top the data presented. This is causing the figure very crowded. Please remove such unnecessary box and keep it simple.

**Do you want your identity to be public for this peer review?** For information about this choice, including consent withdrawal, please see our Privacy Policy

Reviewer #2: **Yes:** Ranjan Tamuli

---

## [Author Response · Author response to Decision Letter 2]

15 Feb 2026

Response to the comments:

Comments:

The manuscript titled “Genetic and pharmacologic inhibition of calcineurin reduces biofilm formation by the pathogenic fungus Trichosporon asahii in an in vivo silkworm infection model” by Matsumoto has been revised by taking care of the suggestions made in the initial version. The manuscript has been considerably improved. However, I recommend for additional corrections as suggested below. The strains used must be described well in the Materials and Methods section, particularly, the complementation strain generation procedure used for generating the complemented strains. In addition, please check the language thoroughly in the final version.

1. Page 5: Line 72: “In the present study, we found that while the cnb1 gene-deficient T. asahii mutant.” In the last paragraph of the introduction, only the work done on cnb1 has been described. Please also described the work done on cna1, which is now described in the results section of the revised manuscript.

According to the reviewer’s comment, we added the sentences in the Introduction section of the revised manuscript (Page 5, line 72-74).

[Page 5, lines 72-74]

In the present study, we found that while the mutants deficient for cna1 and cnb1 genes formed biofilms in vitro, in vivo biofilm formation was reduced in the catheter-inserted silkworm infection model.

2. Page 6: Lines 87-94: Please change “T. asahii cultures” as “Generation of the T. asahii strains and culture condition”. In addition, please briefly describe the mutants and how the complement strains were generated. The author cited reference by Matsumoto et al 2022; however, the complement strain generation was not described. Please describe the procedure to generate the complement strain well.

According to the reviewer’s comment, we added the method to generate the complement strains in the Materials and Methods section of the revised manuscript (Page 6, lines 91-95).

[Page 6, lines 91-95]

The cna1 or cnb1 gene-deficient strains were generated by replacing each target gene using 5’-UTR (cna1) -NAT1-3’-UTR (cna1) or 5’-UTR (cnb1) -NAT1-3’-UTR (cnb1) fragments [22]. Complemented strains were generated by reintroducing each target gene using 5’-UTR (cna1) -cna1-hph-3’-UTR (cna1) or 5’-UTR (cnb1) -cnb1-hph-3’-UTR (cnb1) fragments [22].

3. Page 11: lines 182-183: “Fluorescence intensity of the cna1 gene- and the cnb1 gene-deficient mutants were not decreased compared...........”, please correct as “Fluorescence intensity of the mutants deficient for the cna1 and cnb1 genes were not decreased compared....”

Following the reviewer’s comment, the sentence was changed (Page 11, lines 186-188).

[Page 11, lines 186-188]

Fluorescence intensity of the mutants deficient for the cna1 and cnb1 genes were not decreased compared with that of the parent strain (Fig. 2).

4. Page 11: lines 187-188 “Fig. 1. Adhesion and biofilm formation by cna1 gene- and cnb1 gene-deficient mutants of T. asahii in vitro.”, please correct as “Fig. 1. Adhesion and biofilm formation in the T. asahii cna1 and cnb1 mutants.”

According to the reviewer’s comment, the sentence was changed (Page 11, line 191).

[Page 11, line 191]

Adhesion and biofilm formation in the T. asahii cna1 and cnb1 gene-deficient mutants in vitro.

5. Page 17: Lines 278-279: “The cna1 gene- and cnb1 gene-deficient mutant exhibits sensitivity to several stresses, including membrane damage (sodium dodecyl sulfate), cell wall stress (Congo red), oxidative stress (H₂O₂), and endoplasmic reticulum stress (tunicamycin and dithiothreitol) [22].”, please correct as “The cna1 and cnb1 deficient mutant exhibited sensitivity to membrane damaging agent sodium dodecyl sulfate (SDS), cell wall stress induced by Congo red (CR), oxidative stress mediated by hydrogen peroxide (H₂O₂), and endoplasmic reticulum stress caused by tunicamycin and dithiothreitol [22].”

Following the reviewer’s comment, the sentence was changed (Page 17, lines 281-284).

[Page 17, lines 281-284]

The cna1 and cnb1 gene-deficient mutants exhibited sensitivity to membrane damaging agent sodium dodecyl sulfate (SDS), cell wall stress induced by Congo red (CR), oxidative stress mediated by hydrogen peroxide (H₂O₂), and endoplasmic reticulum stress caused by tunicamycin and dithiothreitol [22].

6. Page 21: Lines 338-340: Figure 6: The model described only the effect of tacrolimus. However, the author used the cna1 and cnb1 mutants, and the results obtained using these mutants constitute the major part of the manuscript. Therefore, I recommend improving the model by incorporating the results using the mutants.

According to the reviewer’s comment, we changed the model in Figure 6.

7. In several figures, the cna1, cnb1 are shown within a box on top the data presented. This is causing the figure very crowded. Please remove such unnecessary box and keep it simple.

According to the reviewer’s comment, we removed the box in the Figures of the revised manuscript.

---

## [Editor Report · Decision Letter 2]

18 Feb 2026

Genetic and pharmacologic inhibition of calcineurin reduces biofilm formation by the pathogenic fungus Trichosporon asahii in an in vivo silkworm infection model

PONE-D-25-62288R2

Dear Dr. Matsumoto,

We’re pleased to inform you that your manuscript has been judged scientifically suitable for publication and will be formally accepted for publication once it meets all outstanding technical requirements.

Kind regards,

Katherine A. Borkovich, Ph.D.

Academic Editor

PLOS One
---

## [Editor Report · Acceptance letter]

PONE-D-25-62288R2

PLOS One

Dear Dr. Matsumoto,

I'm pleased to inform you that your manuscript has been deemed suitable for publication in PLOS One. Congratulations! Your manuscript is now being handed over to our production team.

Kind regards,

on behalf of

Dr. Katherine A. Borkovich

Academic Editor

PLOS One